# Single-cell analysis reveals dynamics of human B cell differentiation and identifies novel B and antibody-secreting cell intermediates

**Niels JM Verstegen[1,2†], Sabrina Pollastro[1†], Peter-Paul A Unger[1], Casper Marsman[1], George Elias[1], Tineke Jorritsma[1], Marij Streutker[1], Kevin Bassler[3], Kristian Haendler[3,4], Theo Rispens[1], Joachim L Schultze[3,4], Anja ten Brinke[1], Marc Beyer[3,4,5], S Marieke van Ham[1,6]***

[1]Department of Immunopathology, Sanquin Research and Landsteiner Laboratory, Amsterdam UMC, University of Amsterdam, Amsterdam, Netherlands; [2]Synthetic Systems Biology and Nuclear Organization, Swammerdam Institute for Life Sciences, University of Amsterdam, Amsterdam, Netherlands; [3]Genomics and Immunoregulation, University of Bonn, Bonn, Germany; [4]Platform for Single Cell Genomics and Epigenomics, German Center for Neurodegenerative Diseases (DZNE), University of Bonn, Bonn, Germany; [5]Immunogenomics & Neurodegeneration, German Center for Neurodegenerative Diseases, Bonn, Germany; [6]Swammerdam Institute for Life Sciences, University of Amsterdam, Amsterdam, Netherlands

**\*For correspondence:** m.vanham@sanquin.nl

[†]These authors contributed equally to this work

**Competing interest:** The authors declare that no competing interests exist.

**Abstract** Differentiation of B cells into antibody-secreting cells (ASCs) is a key process to generate protective humoral immunity. A detailed understanding of the cues controlling ASC differentiation is important to devise strategies to modulate antibody formation. Here, we dissected differentiation trajectories of human naive B cells into ASCs using single-cell RNA sequencing. By comparing transcriptomes of B cells at different stages of differentiation from an in vitro model with ex vivo B cells and ASCs, we uncovered a novel pre-ASC population present ex vivo in lymphoid tissues. For the first time, a germinal-center-like population is identified in vitro from human naive B cells and possibly progresses into a memory B cell population through an alternative route of differentiation, thus recapitulating in vivo human GC reactions. Our work allows further detailed characterization of human B cell differentiation into ASCs or memory B cells in both healthy and diseased conditions.

## Editor's evaluation

To recapitulate B cell differentiation process in vitro, the authors established an in vitro system to identify a cluster and performed extensive analyses to demonstrate that the cluster mimics human germinal center antibody-secreting cells (ASCs). They provide stepwise trajectories of plasma cell differentiation from human naive B cells upon stimulation with CD40 ligands, IL-4, and IL-21. Since intermediate clusters of cells show features of germinal center B cells, the authors propose novel intermediate stages of B cells before a complete differentiation into plasma cells. This study is valuable in the differentiation of primary naive B cells into ASC ex vivo and may interest immunologists with an emphasis in B cell biology as it provides an in-depth description of the B cell differentiation pathways.

## Introduction

Protection from invading pathogens is, in part, provided by antibody-secreting cells (ASCs) that secrete high-affinity class-switched antibodies. Antibody formation, however, can also be highly undesired. This is the case for detrimental autoantibodies in autoimmune disorders, for alloantibodies that react with donor blood cells after a blood transfusion, or for antibodies that are formed against therapeutic biologics and that interfere with their therapeutic efficacy. Plasma cells secreting high-affinity antibodies are generated from naive B cells that, upon foreign antigen recognition by the antigen-specific B cell receptor (BCR), get activated and after CD4 T cell help can start a germinal center (GC) reaction. Here, B cells differentiate into GC B cells and begin to cycle between the GC dark zone (DZ) and the GC light zone (LZ). In the former, GC B cells undergo several rounds of cell cycle and receptor editing processes, such as somatic hypermutation and class-switching. In the latter, B cells need to reacquire antigen and present antigenic peptides to cognate antigen-specific CD4 T follicular helper (Tfh) cells to receive the required co-stimulatory, survival, and differentiation signals (*Bryant et al., 2007*; *King and Mohrs, 2009*; *King et al., 2008*; *Reinhardt et al., 2009*; *Yusuf et al., 2010*; *Glatman Zaretsky et al., 2009*). After several rounds of DZ to LZ cycling, this process results in the selection of high-affinity, class-switched B cells (reviewed in *Victora and Nussenzweig, 2022*). While early memory B cell and early plasmablast formation occur before GC establishment (*Glaros et al., 2021*), the GC reactions give rise to a later stage, higher affinity memory B cell formation followed by the eventual formation of high-affinity, ASCs (plasmablasts and plasma cells) (*Weisel et al., 2016*). Migration of plasma cells to the bone marrow and incorporation into bone marrow survival niches establishes long-lived plasma cells that may secrete antibodies for decades (*Belnoue et al., 2008*; *Cassese et al., 2003*; *Radbruch et al., 2006*).

The transition of B cell into ASC is tightly regulated on the molecular level by a complex network of transcription factors that control the switch from the B cell stage to the ASC stage (reviewed in *Verstegen et al., 2021*). The ASC-specific transcription factors IRF4, XBP1, and BLIMP1 are upregulated only once the B cell-specific transcription factors PAX5, IRF8, and BACH2 are downregulated. These transcriptional changes are then followed by surface expression of stage-specific cellular markers, such as CD27, CD38, and CD138 on ASCs, with simultaneous downregulation of the B cell markers CD19 and CD20 (*Loken et al., 1987*; *Sanderson et al., 1989*; *Tangye et al., 1998*; *Terstappen et al., 1990*; *Victora et al., 2012*).

Although these transcription factors associated with plasma cell differentiation and plasma cell markers have been well-studied, the differentiation trajectories of naive B cells into short-lived and long-lived ASCs and the regulators that control transition at the different stages of human B cell differentiation remain largely unknown. Bulk RNA sequencing on sorted B cells ex vivo has identified differences between naive B cells, GC B cells, memory B cells, plasmablasts, and plasma cells (*Halliley et al., 2015*; *Minnich et al., 2016*; *Tarte et al., 2003*), but does not allow delineation of the actual differentiation process and the various stages through which B cells transition to becoming ASCs. The development of single-cell sequencing techniques circumvents this problem and allows for the discovery of new cellular subsets and cells that are transitioning from one cellular differentiation state to another. To date, single-cell sequencing has been used, among others, to identify B cell subsets in the immune landscape of tumor-infiltrating lymphocytes (*Chung et al., 2017*), delineate malignant B cells in follicular lymphomas (*Andor et al., 2019*), and identify transiting bone marrow progenitor cell populations that commit to the B cell lineage (*Alberti-Servera et al., 2017*; *Miyai et al., 2018*). In addition, software to reconstruct B cell receptor sequences from single-cell RNA sequencing data has been developed in recent years (*Canzar et al., 2016*; *Lindeman et al., 2018*; *Rizzetto et al., 2018*; *Upadhyay et al., 2018*). In-depth characterization of the differentiation process of mature naive B cells into ASCs has been investigated by focusing on parts of this differentiation process, for example, GC LZ to DZ migration (*Dominguez-Sola et al., 2015*; *Milpied et al., 2018*; *Sander et al., 2015*). Although this contributed to new insights into GC reactions, it remains unclear how differentiation into plasma cells occurs and is regulated. More insight into this differentiation process might uncover targets to regulate B cell differentiation into ASC at an early stage and to intervene in the generation of undesired antibodies in disease.

We previously established a minimalistic in vitro culture system that uniquely supports efficient in vitro differentiation of human naive B cells into CD27++CD38++ ASCs while also maintaining high cell numbers along the differentiation stages (*Unger et al., 2021*). Here, we applied in-depth single-cell

transcriptomics, B cell receptor reconstruction and trajectory inference on the in vitro-generated ASCs and combined analyses with data from publicly available single-cell datasets of ex vivo obtained B cells and ASCs from different human tissues. Our data demonstrate that in vitro-generated ASCs are highly comparable to their ex vivo counterparts. The differentiation trajectories uncover a hitherto unknown pre-ASC cellular stage that is detected also ex vivo in lymphoid tissue with active ASC differentiation processes. For the first time, GC-like B cells are identified in vitro from human naive B cells and possibly progress through an alternative route of differentiation into memory B cells. In addition to known regulatory transcription factors and cell surface markers, potential novel transcriptional regulators and plasma membrane markers are identified that may control or, respectively, mark the process of human plasma cell differentiation.

## Results

### Single-cell transcriptomic analysis of in vitro differentiated antibody-secreting cells

We previously described a minimalistic in vitro system that effectively differentiates human naive B cells into ASCs while also maintaining high cell numbers at the various stages of B cell differentiation (*Unger et al., 2021*). Briefly, human naive B cells (CD19$^+$CD27$^-$IgG$^-$IgD$^+$) are sorted from healthy donor peripheral blood mononucleated cells (PBMCs) and cultured on a feeder layer of human CD40L-expressing mouse fibroblasts and restimulated after 6 days. Cytokines typically expressed by follicular T cells (Tfh) (IL-21, IL-4) are added to mimic Tfh help for B cell differentiation. Restimulation with CD40 costimulation and Tfh cytokines in vitro was demonstrated to be needed to drive efficient ASC differentiation, in line with in vivo GC reactions (*Figure 1a*). Triggering of the IgM BCR with soluble anti-IgM antibodies or IgM-coated beads did not contribute to enhanced ASC differentiation in the minimalistic cultures (*Unger et al., 2021*). To assess the transition of naive B cells (CD27$^-$CD38$^-$) into ASCs (CD27$^{++}$CD38$^{++}$), cells from the four CD27/CD38 quadrants were sorted on day 11 and the expression of master regulators of B cells to ASC differentiation was analyzed (*Figure 1b*; *Verstegen et al., 2021*). CD27$^-$CD38$^-$ cells gradually lose expression of B cell signature genes *PAX5* and *BACH2* upon acquisition of CD27 and/or CD38 expression with CD27$^{++}$CD38$^{++}$ cells showing the most down-regulated expression of *PAX5* and *BACH2* and upregulation of ASC signature genes *IRF4*, *XBP1*, and *PRDM1* (*Figure 1c–g*). In line with this, CRISPR-Cas9-mediated knockout of *PRDM1* on day 3 strongly suppressed the formation of CD27$^+$CD38$^+$ cells on day 11 compared to control and *CD19* knockout (*Figure 1h*).

Since the minimalistic in vitro system efficiently drives differentiation of primary human naive B cells into ASCs and follows the in vivo observed transitions of the master regulators of transcription, the dynamics of human naive B cell differentiation were investigated in detail by single-cell RNA sequencing. Cultured cells were single-cell sorted on day 11 based on the expression of CD27 and CD38 to obtain a faithful representation of cells at varying stages of B cell to ASC differentiation. Next, cells were processed for single-cell RNA sequencing using the SMARTseq2 method. After raw data processing and quality control, a total of 275 (out of 382 sorted cells) high-quality cells that together express 17,716 genes were included in the final dataset (*Figure 1—figure supplement 1a–e*). Cells were assigned to one of three phases of the cell cycle (G1, G2/M, S) as determined by scores calculated based on the expression of G2/M and S phase genes (*Figure 1—figure supplement 1f*). As the cell cycle is an essential component of GC reactions, data were analyzed with and without regressing cell-cycle heterogeneity. When the cell cycle was not regressed out, cells were clustered based on cell-cycle scores, and most differentially expressed genes were related to the cell cycle (*Figure 1—figure supplement 1g*). Cell-cycle heterogeneity was regressed out in the final dataset to primarily focus our analysis on factors influencing B cells differentiation apart from the cell cycle (*Figure 1—figure supplement 1h and i*).

Unsupervised hierarchical clustering and visualization with Uniform Manifold Approximation and Projection (UMAP) identified five clusters of differentiating cells (*Figure 1i*, *Figure 1—figure supplement 1j*). Clusters 4 and 5 showed a clear ASC gene signature with a prominent expression of known ASC-specific genes *IRF4*, *XBP1*, and *PRDM1* (*Figure 1j–l*). Of note, 99% of CD27$^+$CD38$^+$ double-positive sorted cells were represented in clusters 4 and 5 (*Figure 1—figure supplement 2a and b*). The other three clusters (clusters 1–3) still showed a pronounced B cell signature with higher

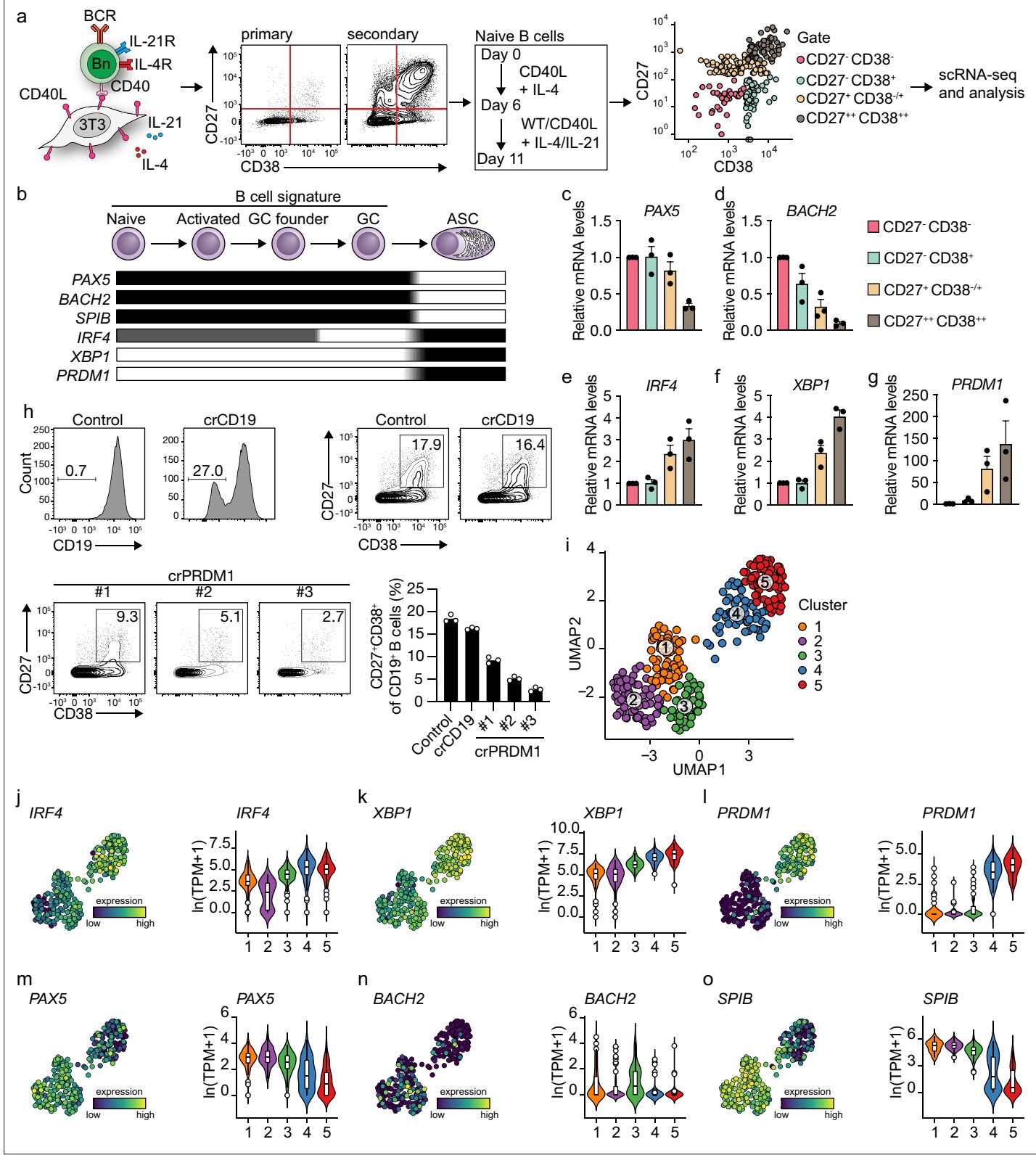

**Figure 1.** Unbiased analysis of in vitro-induced human naive B cell differentiation by scRNA-seq reveals five transcriptionally distinct B cell clusters. (**a**) Overview of the scRNA-seq experiment in short human naive B cells was isolated and cultured with a human CD40L-expressing mouse fibroblast (3T3) cell line and recombinant IL-4. Secondary cultures were initiated on day 6 using 9:1 wild type (WT) 3T3 to CD40L-expressing 3T3 cells and IL-4/IL-21 as this was necessary to induce substantial differentiation into the CD27++CD38++ ASC subset. On day 11, 384 cells were single-cell sorted based on the

*Figure 1 continued on next page*

*Figure 1 continued*

expression of CD27 and CD38 and sequenced using the Smart-seq2 method. (**b**) Overview of the cellular stages and important transcription factors involved in B cells differentiation from naive to antibody-secreting cell (GC, germinal center; ASC, antibody-secreting cell). (**c–g**) Expression of *PAX5* (**c**), *BACH2* (**d**), *IRF4* (**e**), *XBP1* (**f**), and *PRMD1* (**g**) mRNA in sorted populations was analyzed by qPCR and related to levels present in CD27⁻CD38⁻ cells. Each data point represents the mean of an individual experiment (n = 3) with triplicate measurements. Mean values are represented as bars. (**h**) FACS analysis of surface CD19 levels (top left) and ASC differentiation (top right) in cultured primary human B cells expressing the indicated control or CD19-targeting RNP. FACS analysis of ASC differentiation (bottom left) and quantification (bottom right) in cultured primary human B cells expressing the indicated control or three different *PRDM1*-targeting RNPs. Representative of three independent experiments with triplicate measurements. (**i**) Uniform Manifold Approximation and Projection (UMAP) projection of single-cell transcriptomes of in vitro differentiated human naive B cells (276 high-quality cells). Each point represents one cell, and colors indicate graph-based cluster assignments. (**j–o**) UMAP projection as in (**i**) colored by the transcriptional regulators *IRF4* (**j**), *XBP1* (**k**) and *PRMD1* (**l**), *PAX5* (**m**), *BACH2* (**n**), *SPIB* (**o**), which are important in B cell differentiation (left graph of each panel), along with corresponding distribution of average expression levels (ln(TPM+1)) across the B cell clusters (1, 2, and 3) and the ASC clusters (4 and 5) (right graph of each panel).

The online version of this article includes the following figure supplement(s) for figure 1:

**Figure supplement 1.** scRNA-seq analysis of differentiating human B cells.

**Figure supplement 2.** Subset distribution of differentiating primary human B cells.

expression of genes known to repress the ASCs state such as *PAX5*, *BACH2,* and *SPIB* (*Figure 1m–o*). Thus, upon 11 days of culture with CD40 costimulation and IL-21/IL-4, human naive B cells differentiate into distinct B cells and ASC clusters based on overall gene signatures.

## In vitro-generated ASCs are highly comparable with ex vivo obtained ASCs from different human tissues

To further characterize in vitro-generated ASCs, transcribed immunoglobulin genes were reconstructed from the single-cell transcriptomic data using *BraCeR,* and transcription levels and properties of immunoglobulins genes were compared between B cell and ASC clusters. A functional B cell receptor heavy chain rearrangement was successfully reconstructed for 255 out of 276 cells analyzed. Expression of the reconstructed B cell receptor (BCR) genes in clusters 4 and 5 was up to 10 times higher compared to clusters 1–3, demonstrating a very high transcriptional activity of immunoglobulin genes in the ASC clusters (*Figure 2a*). Isotype analysis of the reconstructed BCR heavy chains revealed a homogeneous distribution of isotypes in the different clusters (*Figure 2b*, *Figure 2—figure supplement 1a and b*). When combining data for B cell and ASC clusters, the percentage of cells expressing immunoglobulin of the IgM isotype was lower in clusters 4 and 5 compared to clusters 1–3 (clusters 4 and 5: 36.6% vs. clusters 1–3: 50.3%; p-value<0.01), while the opposite was true for expression of immunoglobulins of the IgG1 isotype (cluster 4 and 5: 43.7% vs. cluster 1–3: 24.6%; p-value <0.001). In addition, 87% (226/255) of all the reconstructed BCR heavy genes were identical to the germline sequence, that is, not somatically hypermutated, independently of their cluster origin (*Figure 2c*). Thus, in vitro-generated ASCs express a high load of, mostly, IgG1 class-switched but no significant mutated immunoglobulin genes.

To assess how the transcriptional profile of the in vitro-generated ASCs was representative of the in vivo situation, the specific gene signature of the in vitro-generated ASCs was first identified by performing differential expression analysis in the ASC clusters 4 and 5 compared to the B cell clusters 1–3 (*Figure 2d*). A total of 3858 genes were differentially expressed in the ASC clusters, with 1406 overexpressed among which *PRDM1* and *XBP1* were top hits. Next, publicly available scRNA-seq datasets from bone marrow from the Human Cell Atlas Data Coordination Portal 'Census of Immune Cells' project (https://data.humancellatlas.org/explore/projects/cc95ff89-2e68-4a08-a234-480eca21ce79; *Figure 2—figure supplement 2*), human peripheral blood (*Rizzetto et al., 2018*; *Figure 2—figure supplement 3*)**,** and tonsils (*Attaf et al., 2020*; *Figure 2—figure supplement 4*) were used to perform differential expression analysis in which each tissue-specific ASC cluster was compared to its tissue-specific B cell counterpart. A total of 7653, 1079, and 1860 genes were upregulated in ex vivo obtained ASCs from bone marrow, peripheral blood, and tonsils, respectively (*Figure 2e–g*). Analysis of the overlap between the identified upregulated genes selected in in vitro and ex vivo data revealed that out of the 1406 upregulated genes found in in vitro-generated ASCs, 1113 (79%) were shared with at least one ex vivo ASC population, of which 349 (25%) were shared with all populations (*Figure 2h and i*). The remaining 293 (21%) genes were uniquely upregulated in in vitro-generated

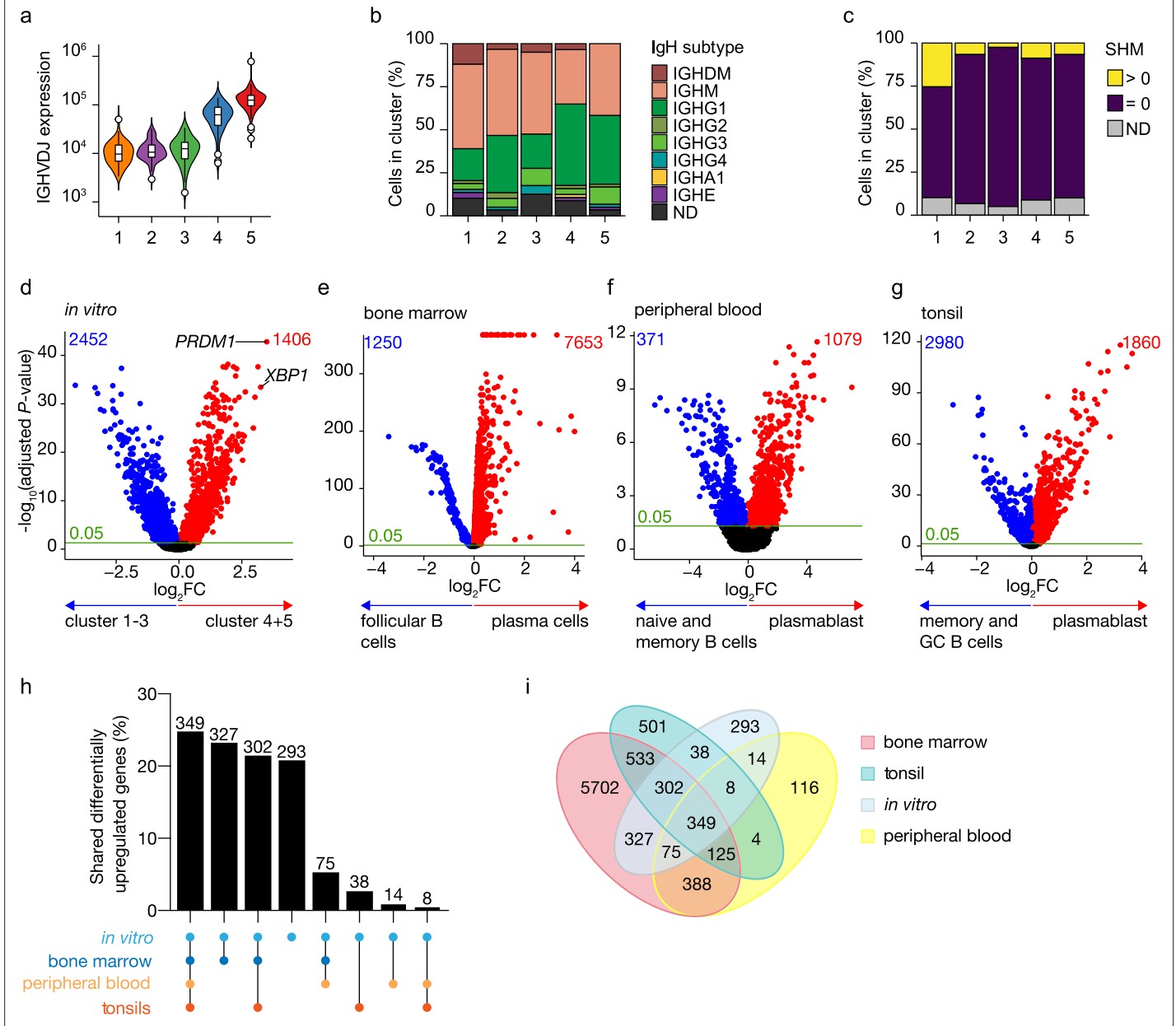

**Figure 2.** Identification of antibody-secreting cells with substantial gene expression overlap as compared to ex vivo-derived plasmablast/plasma cells among the in vitro-differentiated human naive B cells. (**a**) Violin plot of the reconstructed B cell receptor immunoglobulins gene expression as determined by BraCeR within each B cell cluster. (**b**) Stacked bars denotes the frequency of isotype analysis of the reconstructed immunoglobulin heavy chain within each cluster. (**c**) Stacked bars showing the percentage of cells with a given B cell receptor mutation count within each B cell cluster. (**d–g**) Volcano plot depicting significantly (adjusted p-value <0.05) down- and upregulated genes. (**d**) In vitro-generated antibody-secreting cells (clusters 4 and 5), ex vivo bone marrow-derived plasma cells (**e**), ex vivo peripheral blood-derived plasmablast (**f**), and ex vivo tonsil-derived plasmablast/plasma cells (**g**) were compared to each specific B cell counterparts. (**h, i**) UpSetR plot (**h**) and Venn diagram (**i**) depict the intersection among all upregulated genes identified in in vitro and ex vivo antibody-secreting cells.

The online version of this article includes the following figure supplement(s) for figure 2:

**Figure supplement 1.** The distribution of heavy immunoglobulin gene in differentiating primary human B cells.

**Figure supplement 2.** Selection of bone marrow-derived mature B cells and antibody-secreting cells.

**Figure supplement 3.** Gene expression characteristics of peripheral blood naive B cells, memory B cells, and plasmablast.

**Figure supplement 4.** Gene expression characteristics of tonsillar memory B cells, germinal center (GC) B cells, and plasmablast/plasma cells.

ASCs, indicating that these genes might be linked to in vitro culture rather than be general for ASCs. Additionally, ex vivo ASC populations also possess unique differentially expressed genes that are not shared with other ex vivo or in vitro-generated ASC populations (*Figure 2i*). These findings indicate that in vivo ASCs carry specific, possibly tissue-related, gene signatures in addition to the 'core ASC signature' shared among ASCs. Taken together these data show that the in vitro-generated ASCs have highly similar transcriptional profiles compared to ex vivo-obtained ASCs.

## A novel B cell to ASC intermediate cellular stage is identified as a precursor of terminally differentiated ASCs

In our dataset, cells with a prominent ASC gene signature were separated into two distinct transcriptional clusters. The cells in these clusters also show a slightly different expression of the ASCs phenotypic markers CD27 and CD38, with cells in cluster 5 being predominantly $CD27^{++}CD38^{++}$, while cluster 4 also includes $CD27^{-}CD38^{+}$ cells (*Figure 1—figure supplement 2a*). When performing clustering analysis on the dataset without cell-cycle regression, again cells from clusters 4 and 5 separated from the B cells clusters. Interestingly, the two former ASC clusters now separated into two new clusters, whereby the cell cycle stage became the major common denominator between the two clusters (*Figure 1—figure supplement 1j*). This shows that original clusters 4 and 5 contain both cells in the cell cycle and cells that are out of the cell cycle. To further investigate the differences between cells in clusters 4 and 5, the similarity and the enrichment score for ASC gene signature were analyzed for both clusters (*Figure 3a and b*). Cells in cluster 4 scored lower than cells in cluster 5 for both ASC similarity and enrichment, indicating a less dominant ASC gene signature in this cluster. Differential expression analysis between clusters 4 and 5 identified a total of 1076 differentially expressed genes, with 114 and 962 genes up- and downregulated in cluster 5, respectively (*Figure 3c*). Network analysis of Gene Set Enrichment Analysis (GSEA) results for the differentially expressed genes revealed a predominant enrichment for terms concerning cytoskeleton organization, including supramolecular fiber organization, immune cell receptor signaling, and metabolism of steroids (*Figure 3d*, *Figure 3—figure supplement 1*). Of note, among the enriched terms concerning cytoskeleton organization, we found '*Arp2/3 complex-mediated actin nucleation*,' recently shown to be important in immune synapses formation (*Bolger-Munro et al., 2019*; *Roper et al., 2019*), BCR signaling and B cell activation, together with other terms involved in antigen uptake and processing (*Figure 3—figure supplement 1*). The majority of the processes indicated by the predominantly enriched terms were suppressed in cluster 5 compared to cluster 4, while specifically activated processes in cluster 5 were '*protein N-linked glycosylation*' and '*retrograde protein transport*' (*Figure 3e*). Thus, cells in cluster 4 are still active in antigen presentation and processing, BCR signaling, and fatty acid metabolism, while such processes are being shut down in cells from cluster 5 that conversely upregulate genes involved in protein glycosylation and protein transport.

To assess whether cluster 4 is a precursor population of cluster 5, we applied two algorithms for analyzing respectively the connectivity and the dynamics between clusters in scRNA-seq datasets, namely, partition-based graph abstraction (PAGA) (*Wolf et al., 2019*) and RNA velocity (*Bergen et al., 2020*). Projection of the RNA velocity vectors on the UMAP representation revealed an area of increased velocity in cluster 4 in the direction of cluster 5, indicating that cells in cluster 4 are rapidly transitioning toward the transcriptomic signature of cells in cluster 5 (*Figure 3f*). At the top-right extremity of cluster 5, where most of the non-cycling cells are located, no velocity vectors are projected. This implies that no subsequent cellular stage could be appointed, suggesting an endpoint of differentiation here. In line with this, results of PAGA showed increased connectivity among clusters 4 and 5 (*Figure 3g*). Interestingly, only cluster 4 and not cluster 5 was shown to be connected with all three B cell clusters, with cluster 3 being the most pronounced connection followed by cluster 1. This indicates that cluster 4 is more similar to the B cell clusters than cluster 5.

To confirm that cluster 4 represents a previously overseen intermediate B cell to ASCs population, we obtain a set of hallmark genes for cluster 4 and tested them on the publicly available scRNA-seq dataset from bone marrow, tonsil, and peripheral blood (*Figure 3h*). In the bone marrow dataset, the highest similarity score with the cluster 4 hallmark genes was observed in the cells connecting naive B cells to plasma cells. Interestingly, the GC B cells in the tonsil dataset, both non-cycling LZ and cycling DZ GC B cells, showed the highest similarity score with cluster 4 cells, while in the peripheral blood dataset, few cells in the naive/memory cluster showed increased similarity. Of note, when performing

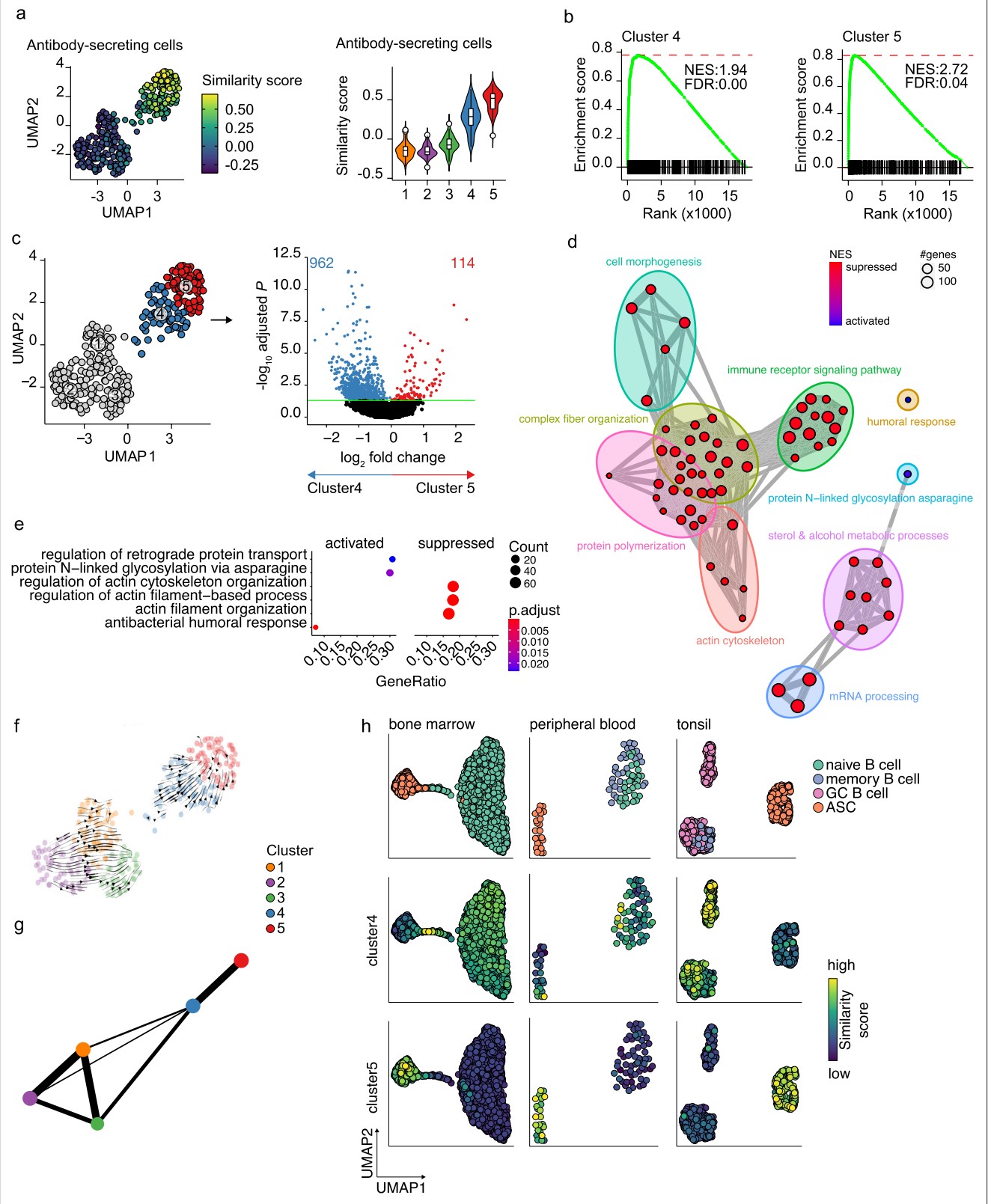

**Figure 3.** Pre- and terminally differentiated antibody-secreting cells (ASCs) separated in different clusters. (**a**) Uniform Manifold Approximation and Projection (UMAP) projection colored by ASC similarity score (left), along with corresponding distribution of the similar score within each cluster (right). (**b**) Gene Set Enrichment Analysis (GSEA) enrichment plots for ASC gene signature in cells from clusters 4 (left) and 5 (right) compared to the other clusters (NES, normalized enrichment score; FDR, false discovery rate). (**c**) UMAP projection as in *Figure 1i* colored by clusters 4 and 5 membership

*Figure 3 continued on next page*

*Figure 3 continued*

(left) and volcano plot depicting significantly (adjusted p-value<0.05) down- and upregulated genes in cluster 5 compared to cluster 4. (**d**) Functional grouping network diagram of GSEA comparing clusters 4 and 5. The dot size and the dot color represent the number of genes in the pathway and the NES, respectively. (**e**) The top 3 most significantly differentially activated pathways as determined by GO enrichment analysis comparing clusters 4 and 5. The dot size and the dot color represent the number of genes in the pathway and the adjusted p-value, respectively. (**f**) Velocyto force field showing the average differentiation trajectories (velocity) for cells located in different parts of the UMAP plot. Arrow size conveys the strength of predicted directionality. (**g**) PAGA graph showing the connectivity between the clusters. Each node corresponds to each of the clusters identified using Seurat. The most probable path connecting the clusters is plotted with thicker edges. (**h**) UMAP projection of ex vivo B cells and ASC derived from bone marrow, peripheral blood, and tonsil colored by cluster 4 and 5 similarity score.

The online version of this article includes the following figure supplement(s) for figure 3:

**Figure supplement 1.** Enrichment map of Gene Set Enrichment Analysis comparing clusters 4 and 5.

the same analysis with the hallmark genes from cluster 5, the highest similarity was observed in each tissue-specific ASC population, as expected.

Thus, taken together these results show that cluster 5 represents a terminally differentiated ASC population and that cluster 4 represents a novel B cell to ASC intermediate cellular stage, a precursor of terminally differentiated ASCs. This population can also be found ex vivo, mainly in lymphoid tissues with active ASC differentiation.

## Differentiating B cells transit through a GC-like and early memory phenotype

To analyze the earlier stage of naive B cell to ASC differentiation, differential expression analysis among the three B cell clusters was performed to identify cluster-specific gene signatures (*Figure 4a*). Interestingly, genes that mark a GC B cell phenotype, such as *IRF8*, *BCL6*, *CD22*, and *CD83*, were among the top 30 differentially expressed genes in cluster 2 (*Basso and Dalla-Favera, 2010*; *Meyer et al., 2021*; *Victora et al., 2012*; *Wang et al., 2019*). When performing GSEA using hallmark gene sets on the differentially expressed genes in cluster 2, MYC (*Calado et al., 2012*; *Dominguez-Sola et al., 2012*) and E2F (*Béguelin et al., 2017*) targets, known to promote GC B cell differentiation, were the top 3 significantly enriched gene sets and, in fact, *MYC* was expressed at its highest in cluster 2 compared to the other clusters (*Figure 4b and c*). Finally, the similarity score with ex vivo-sorted human GC B cells was the highest in cluster 2, followed by cluster 3 (*Figure 4d*). Thus, we concluded that cells in cluster 2 represent GC-like B cells.

For cells in cluster 3, no definite phenotypic signature could be appointed based on the cluster-specific gene signature. However, given the intermediate similarity with both GC B cells and ASC gene signatures (*Figures 3a and 4b*) and the higher connectivity indicated by PAGA with the pre-ASCs cluster 4 compared to the other B cell cluster, we hypothesize that cluster 3 might represent a post-GC population primed to become ASCs.

Among the top differentially expressed genes of cluster 1, we found *BANK1*, recently identified as a marker of a pre-memory B cell stage (*Holmes et al., 2020*) together with *CCR6*, which was also expressed at the highest in cluster 1 (*Figure 4e and f*). Of note, *MZB1*, identified in the same study as a marker of pre-ASCs, is expressed at its lowest in cluster 1 (*Figure 4g*). Interestingly, *ITGAX* (CD11c) was indicated as differentially expressed in cluster 1 (*Figure 4h*). This gene defines a heterogeneous population of, mostly, memory B cells initially discovered in the context of aging and autoimmune disease, called atypical memory B cells or age-associated B cells (ABCs) that are thought to be generated through an alternative extra-follicular pathway (*Sutton et al., 2021*; *Wang et al., 2018*). Likewise, the expression of other genes associated with the CD11c+ phenotype, such as *SIGLEC6* and *FCRL5*, was increased in cluster 1 (*Figure 4i and j*) while *CR2* (CD21), *TBX21* (T-bet), *FCRL4*, and *CXCR3* were not expressed in our dataset (data not shown). Finally, *HHEX*, recently shown to direct B cells away from the GC reaction into memory differentiation (*Laidlaw et al., 2020*), was also increased in cluster 1 (*Figure 4k*). Together, these data suggest that cells in cluster 1 are committed to becoming memory B cells.

To confirm the above findings of an in vitro B differentiation system in which GC-like B cells are directed toward both a memory and an ASCs cell phenotype, we analyzed the similarity score of all cells in our dataset ordered based on UMAP1 with gene signatures obtained from ex vivo-sorted B cell at different differentiation stages, namely, GC B cell, memory B cells ,and ASCs (*Figure 4l*). As

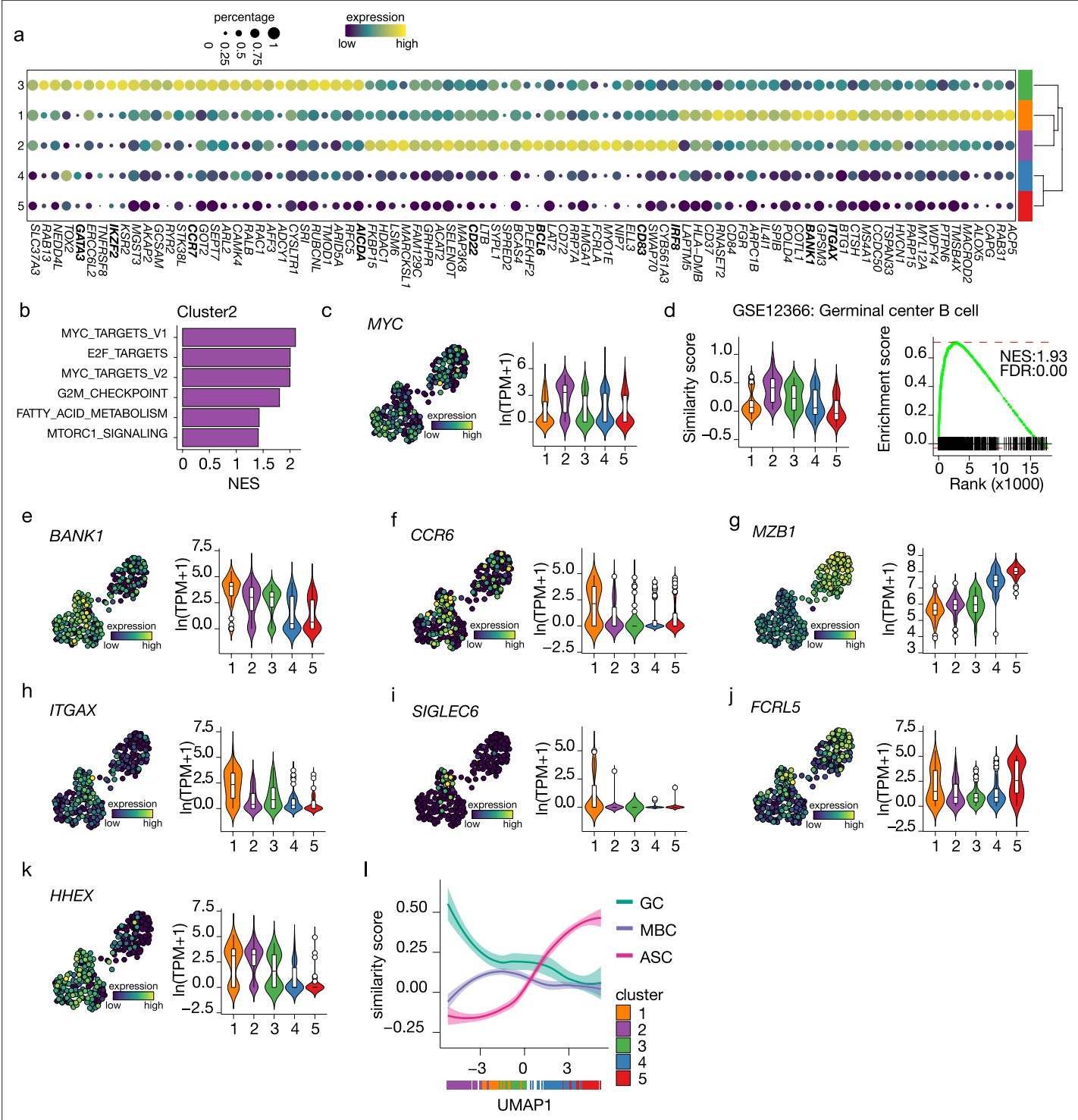

**Figure 4.** Terminal differentiation into antibody-secreting cells (ASCs) starts from germinal center (GC)-like cells. (**a**) Top 30 differentially expressed genes identified in clusters 1–3. The size of the dot encodes the percentage of cells within a cluster, while the color encodes the average gene expression across all cells within a cluster. (**b**) Significantly enriched hallmark gene sets overlapping with the differentially expressed genes identified in cluster 2. (**c**) Uniform Manifold Approximation and Projection (UMAP) projection colored by expression of the transcription factors *MYC* (left), along with corresponding distribution of expression levels (ln(TPM+1)) within each cluster (right). (**d**) Corresponding distribution of the GC similarly score within each cluster (left) and Gene Set Enrichment Analysis (GSEA) enrichment plot (right) for GC B cell gene signature in cells from cluster 2 (NES, normalized enrichment score; FDR, false discovery rate). (**e–k**) UMAP projection colored by expression of the *BANK1* (**e**), *CCR6* (**f**), *MZB1* (**g**), *ITGAX* (**h**), *SIGLEC6* (**i**), *FCRL5* (**j**), *HHEX* (k; left), along with corresponding distribution of expression levels (ln(TPM+1)) within each cluster (right). (**l**) The similarity score of ex vivo germinal center B cell (GC), memory B cell (MBC), and ASC gene sets ordered by UMAP1.

expected, the ASCs similarity score increases following UMAP1-based differentiation, starting at the lowest in cluster 2 and gradually increasing going through clusters 3–5. Conversely, the similarity with GC B cell starts at the highest in cluster 2 and rapidly decreases going toward cluster 1; a slight stagnation is then observed in correspondence to cluster 3, followed by an additional decrease going through clusters 4 and 5. Finally, the similarity with memory B cells starts low with cluster 2, picks up in cluster 1, and decreases again through clusters 3–5. Thus, the differentiation dynamics observed in our in vitro system recapitulate ex vivo naive B cell differentiation processes, including both a memory and an ASC differentiation route.

To conclude, these results indicate that B cells with a GC-like phenotype are identified in vitro from human naive B cells after culture in the presence of IL-4 and IL-21, and CD40L costimulation. Additionally, we find a post-GC B cell population with increased connectivity to the novel B cell to ASCs intermediate and a B cell population with a memory-like phenotype.

## Discussion

In this study, we set out to elucidate different stages of naive human B cell differentiation into ASCs by performing single-cell transcriptomics, B cell receptor reconstruction, and trajectory inference analysis on in vitro-differentiated human naive B cells and integrating these data with ex vivo transcriptomes of isolated B cells and ASC subsets derived from different human tissues. We demonstrated that the in vitro-generated ASC are highly comparable to ex vivo ASCs. This led to the identification of a B cell to ASCs intermediate population still active in antigen presentation and processing that shows transcriptional similarity to B cell populations found ex vivo in lymphoid tissue with active ASCs differentiation processes (like tonsils) and phenotypic resemblance (i.e., intermediate CD27 and CD38 expression, lower expression of *PRDM1*, and higher expression of *PAX5* compared to CD138 plasma blasts) to a pre-ASC population recently identified in the peripheral blood of healthy individuals 6 days post-vaccination (*Sanz et al., 2019*). This population might therefore represent a previously overseen B cell to ASCs intermediate that is challenging to identify in normal steady-state conditions as it only appears transiently during ASC formation. Finally, B cells with a GC-like and a pre-memory phenotype were identified in our in vitro system, thus recapitulating the different naive B differentiation processes observed in vivo.

To the best of our knowledge, this is the first study in which GC-like B cells are identified in vitro from human naive B cells. While a similar culture system is well-established and commonly used in mouse B cell biology (*Gonzalez et al., 2018*; *Nojima et al., 2011*), studies utilizing human B cells are limited in comparison. This finding could help us to decode one important and still unanswered question in B cell biology, that is, which factors control the decisions for a GC-B cell to differentiate into a memory B cell or a plasma cell (*Akkaya et al., 2020*). In our system, we were able to clearly define cells with an ASCs phenotype (clusters 4 and 5) and cells with a gene signature that pointed toward a memory-like B cell phenotype (cluster 1). Surprisingly, the fact that both analyses focused on connection and dynamics between clusters, that is, PAGA and RNA velocity, did not indicate a direct strong connection between the B cells clusters and the ASCs clusters raises the question of whether these two populations are related. In this respect, the in vitro-generated ASCs could be differentiating from a cell population that is lost at the time of our analysis or independently of a GC reaction. Additional research including time-course experiments is needed to further explore this possibility.

In line with this, the higher connectivity observed among the three B cell clusters rather than with the ASCs cluster could indicate the presence of an alternative differentiation wave *in time*. In fact, the GC B cells of cluster 2 might represent the next GC reaction after the one that gave rise to the ASCs of clusters 4 and 5, and that would have generated new ASCs, possibly from cluster 3 if the culture would not be interrupted for analysis. This hypothesis raises the intriguing question of whether the memory cells of cluster 1 are again an alternative end point or the starting point of this second GC reaction. Unfortunately, this is bioinformatically difficult to rule out as currents methods to infer cellular trajectory do require a priori knowledge of the start point of differentiation (*Saelens et al., 2019*), while unbiased analysis of cellular stage transition, such as RNA velocity, can still not be corrected for possible other drivers of cellular stage transition such as cell cycle (*Bergen et al., 2021*).

A concept that is now supported by several studies is that different levels of antigen-affinity and different timing of generation during the GC reaction dictate the fate of a GC B cell to become either a memory B cell or a long-lived plasma cell (*Smith et al., 1997*; *Weisel et al., 2016*). Memory B cells

that arise from GC reactions are an earlier output in the GC reaction than plasma cells and therefore accumulate somewhat less (albeit is mostly still considerable) affinity-increasing SHM (*Weisel et al., 2016*). Conversely, long-lived plasma cells are generated at a later stage of the GC reaction and are equipped with highly affine and highly mutated BCRs. Additionally, mouse studies have highlighted the role of Tfh-produced cytokines such as IL-21 and IL-4 in respectively inducing long-lived plasma cells or memory B cells upon in vivo transfer of in vitro-induced GC-like B cells (*Nojima et al., 2011*). In our in vitro system, IL-21 and IL-4 are provided together, explaining the possibility of having both plasma cell and memory populations at the end of the culture. It would be interesting to further analyze the output of a culture in which only one of the cytokines is provided, and in particular, if the presence of IL-4 only could boost the memory population as already demonstrated in vivo in mice (*Duan et al., 2021*). Furthermore, in vitro mouse studies demonstrated that naive B cells need to be primed in culture with IL-4 to obtain the GC-like phenotype and that naive B cells cultured directly in presence of IL-21 proliferate less and directly differentiate into ASCs. Whether the directly generated ASCs are identical to the ASCs generated via the GC-like intermediate stage still needs to be further investigated.

Interestingly, the metabolism of steroids and cholesterol were among the most enriched terms for the differentially expressed genes between the pre- and terminally differentiated ASCs. This is in line with the concept that the full commitment to antibody production and secretion of terminally differentiated ASCs requires a massive reprogramming of the metabolism not only to provide the molecular 'bricks' for antibody production, that is, amino acids and sugars, but also the cellular machinery to sustain such massive production, that is, ER and Golgi expansion. In this respect, several studies demonstrated that the de novo synthesis of fatty acid is a crucial step in plasma cell formation (*Dufort et al., 2014*; *Fagone et al., 2007*). Additionally, some new pieces of evidence support the hypothesis that antibody secretion and ER stress might be linked to the longevity of plasma cells (*Goldfinger et al., 2011*; *Pengo et al., 2013*). In future studies, we shall focus on the analysis of the changes in metabolic pathways that occur in our in vitro systems. A deeper analysis of the metabolic reprogramming happening at the interchange between the different cellular stages observed in our in vitro system might reveal that metabolism plays an additional layer of control on top of the known transcriptional ones for B cell differentiation.

To conclude, using an in vitro system that accurately recapitulates the in vivo human GC reaction we uncovered a novel intermediate population in the ASC differentiation process and possibly an alternative route of differentiation into memory B cells. Further research is needed to further dissect the regulation of human B cell differentiation processes and in particular the choice between the two differentiation routes in both healthy and diseased conditions.

## Materials and methods
### Cell culture
NIH3T3 WT fibroblast cells (3T3) and human CD40L-expressing 3T3 (*Urashima et al., 1996*) were cultured in IMDM (Lonza) containing 10% FCS (Bodinco), 100 U/ml penicillin (Invitrogen), 100 µg/ml streptomycin (Invitrogen), 2 mM L-glutamine (Invitrogen), 50 µM β-mercaptoethanol (Sigma-Aldrich), and 500 µg/ml G418 (Life Technologies).

### Isolation of peripheral blood B cells from human healthy donors
Buffy coats of healthy human donors were obtained from Sanquin Blood Supply. All healthy donors provided written informed consent following the protocol of the local institutional review board, the Medical Ethics Committee of Sanquin Blood Supply, and conforms to the principles of the Declaration of Helsinki. PBMCs were isolated from buffy coats using a Lymphoprep (Axis-Shield PoC AS) density gradient. Afterward, CD19+ B cells were separated using magnetic Dynabeads (Invitrogen) according to the manufacturer's instructions.

### In vitro naive B cell differentiation cultures
CD40L-expressing 3T3 cells were harvested, irradiated with 30 Gy, and 1 × 10⁴ CD40L-expressing 3T3 cells were seeded in B cell medium (RPMI 1640 [Gibco] without phenol red containing 5% FCS, 100 U/ml penicillin, 100 µg/ml streptomycin, 2 mM L-glutamine, 50 µM β-mercaptoethanol, and

20 µg/ml human apotransferrin [Sigma-Aldrich; depleted for human IgG with protein G sepharose]) in 96-well flat-bottom plates (NUNC) to allow adherence overnight. The next day, $2.5 \times 10^4$ human naive (CD19$^+$CD27$^-$IgG$^-$IgD$^+$) B cells were sorted on a FACS Aria and activated using the irradiated CD40L-expressing 3T3 cells in the presence of IL-4 (100 ng/ml; CellGenix) for 6 days.

After 6 days, activated B cells were collected and co-cultured with $1 \times 10^4$ 9:1 WT 3T3 to CD40L-expressing 3T3 cells that were irradiated and seeded 1 day in advance (as described above), together with IL-4 (100 ng/ml) and IL-21 (50 ng/ml; Invitrogen) for 5 days.

## Flow cytometry and sorting

Cells were first washed with PBS and stained with LIVE/DEAD Fixable Near-IR (Dead cell stain kit, Invitrogen) for 30 min at room temperature in the dark. Then, cells were washed with PBS supplemented with 1% bovine serum albumin. Extracellular staining was performed by incubating the cells for 30 min on ice in the dark with the following antibodies: anti-CD19 (clone SJ25-C1, BD Biosciences), anti-CD27 (clone L128, BD Biosciences; clone O323, eBioscience), anti-CD38 (clone HB7, BD Biosciences), anti-CD138 (clone MI15; BD Biosciences), and anti-IgG (clone G18-145, BD Biosciences; clone MH16-1, Sanquin Reagents). Samples were measured on LSRII and analyzed using FlowJo software (Treestar).

For FACS sorting for single-cell RNA sequencing, samples were stained as described previously with LIVE/DEAD Fixable Near-IR (Invitrogen) and the following antibodies: anti-CD19 (clone SJ25-C1, BD Biosciences), anti-CD27 (clone O323, eBioscience), anti-CD38 (clone HB7, BD Biosciences), and anti-CD138 (clone MI15; BD Biosciences). Sorting was performed on Aria IIIu with FACSDiva software v7 optimized for indexed cell sorting. Single cells were sorted by gating CD27/CD38 subpopulations of the living cell compartment. Cells were sorted in 384-well twin.tec plates (Eppendorf) containing 2.3 µl of lysis buffer per well. Lysis buffer consisted of 2.5 µM oligo-dT30 VN primers, 2.5 µM dNTP mix, 0.2% Triton X-100, and 1 U RNase inhibitor. After sorting, plates were centrifuged, snap-frozen on dry ice, and stored at –80°C until further processing.

## Real-time semi-quantitative RT-PCR

Different B-cell subsets (as indicated) were sorted. After sorting, RT–PCR was performed as described before (*Souwer et al., 2009*). Primers were developed to span exon-intron junctions and then validated (*Supplementary file 1*). Gene expression levels were measured in duplicate reactions for each sample in StepOnePlus (Applied Biosystems, Foster City, CA) using the SYBR green method (Applied Biosystems).

## CRISPR–Cas9-mediated gene deletion of primary B cells

CRISPR-targeting RNA (crRNAs) were designed with the InDelphi (https://indelphi.giffordlab.mit.edu), benchling (https://www.benchling.com), and Integrated DNA Technologies design tools (http://www.idtdna.com). Designed crRNAs were synthesized by Integrated DNA Technologies as Alt-R CRISPR/Cas9 crRNAs. crRNA sequences used in the study were as follows: crCD19, 5′-TCCCTCGGTGGG AGACACGG-3′; crPRDM1#1, 5′-CATTAAAGCCGTCAATGAAG-3′; crPRDM1#2, 5′-TGCTCCCGGGGA GAGTGTGC-3′; crPRDM1#3, 5′- GAAGTGGTGAAGCTCCCCTC-3′.

crRNA and trans-activating CRISPR RNA (tracrRNA; Integrated DNA Technologies) were duplexed by heating at 95°C for 5 min. crRNA–tracrRNA duplexes were mixed with Cas9 protein (TrueCut v2; Thermo Fisher) for 10 min at room temperature to form stable ribonucleoprotein (RNP) complexes. $2.5 \times 10^4$ primary human naive (CD19$^+$CD27$^-$IgG$^-$IgD$^+$) B cells were activated for 3 days with $1 \times 10^4$ irradiated CD40L-expressing 3T3 cells in the presence of IL-21 (50 ng/ml). After 3 days, cells were harvested and resuspended in P3 buffer (Lonza), mixed with RNP complexes, added to nucleofector cuvettes (Lonza), and electroporated with Amaxa 4D Nucleofector (Lonza) using program EH-115. Activated and electroporated B cells were then resuspended in a pre-warmed B cell medium and $2.5 \times 10^4$ cells were plated in wells pre-seeded with $1 \times 10^4$ irradiated CD40L-expressing 3T3 cells in the presence of IL-21 (50 ng/ml). After 3 days, B cells were harvested and put on freshly irradiated CD40L-expressing 3T3 cells in the presence of IL-21 (50 ng/ml) for 5 days.

## Preparation of cDNA libraries and sequencing

Sorted cells were processed using the Smart-seq2 protocol (*Picelli et al., 2014*) adjusted for 384-well plates. In short, 384-well plates containing single cells were thawed and cells were lysed and mRNA

denatured by incubating the plates at 95°C for 3 min and put back on ice. Subsequently, 2.7 µl of first-strand cDNA reagent mix, containing 1x first-strand buffer, 5 mM DTT, 1 M betaine, 14 mM MgCl$_2$, 5U RNase inhibitor, 25U Superscript II Reverse Transcriptase, and 1 µM Template-Switching oligonucleotides in nuclease-free water, was added to each well. Plates were incubated at 42°C for 90 min, 70°C for 15 min, and kept on hold at 4°C. Afterward, 7.5 µl of the pre-amplification mix, consisting of 1x KAPA HiFi HotStart Readymix and 0.12 µM ISPCR primers in nuclease-free water, was added to each well. Plates were then incubated as followed: 98°C for 3 min, 23 cycles of 98°C for 20 s; 67°C for 15 s; and 72°C for 6 min, then at 72°C for 5 min and kept on hold at 4°C. Purification of the resulting cDNA was performed using SeraMag SpeedBeads containing 19% w/v PEG 8,000 and a 1:0.8 ratio of cDNA/beads was used for cDNA precipitation. Purified cDNA was eluted in 14 µl of nuclease-free water. The quality of cDNA was randomly checked in 5% of the wells using the Tapestation High-Sensitivity D5000 assay. For the tagmentation reaction, 0.5 µl of 50–150 pg cDNA was mixed with a 1.5 µl Tagmentation mix containing 1x Tagment DNA buffer and Amplicon Tagment mix (Nextera XT DNA sample preparation kit). Plates were incubated at 55°C for 8 min and kept on hold at 4°C. 0.5 µl Neutralize Tagment buffer was added and incubated at room temperature for 5 min to inactivate Tn5. PCR amplification of adapter-ligated cDNA fragments was performed in a final volume of 5 µl containing the two index primers (Nextera XT Index kit) in a 1:5 dilution and Nextera PCR master mix. Plates were then incubated as follows: 72°C for 3 min, 95°C for 30 s, then 14–15 cycles of 95°C for 10 s; 55°C for 30 s; and 72°C for 30 s, then 72°C for 5 min and kept on hold at 4°C. 384 wells (the entire plate, including negative controls) were pooled in a single 2 ml tube (Eppendorf). Purification of cDNA was performed by adding 400 µl of cDNA in a 1:1 ratio with SeraMag SpeedBeads in 19% w/v PEG 8000 in a 1.5 ml LoBind tube. Beads were washed with 1 ml of 80% ethanol. Purified cDNA was eluted in 200 µl of nuclease-free water. The concentration of the library was measured by Qubit according to the manufacturer's instructions. The size distribution of the library was measured using the Tapestation High-Sensitivity D1000 or D5000 assay. The cDNA library pool was stored at –20°C until ready for sequencing.

cDNA library pool was diluted to 2 nM and prepared for sequencing using the NextSeq 500 High Output Kit v2 (75 cycles) according to the manufacturer's instructions. Sequencing was performed on an Illumina NextSeq 500 instrument.

## Preprocessing of single-cell RNA sequencing data

The quality of the retrieved sequencing data was assessed using FastQC (*Andrews, 2010*). Reads were pseudo-aligned to the human transcriptome (gencode version v27) using kallisto (*Bray et al., 2016*). Quantification of gene expression was obtained by gene-level summarization of transcripts abundance using the R package tximport (*Soneson et al., 2015*), without any scaling (countsFromAbundance option set to 'no'). Counts were then normalized using the transcript per million (TPM) method (*Li et al., 2010*) with the following approach: (a) gene length scaling, (b) removal of all immunoglobulins genes and pseudogenes, and (c) sequencing depth scaling (*Figure 1—figure supplement 1a*). Removal of immunoglobulin genes at this stage prevented the artificial skewing of other genes expressed in the ASC clusters due to the high content of immunoglobulins RNA present in this cell type. The scaled TPM matrix was used for downstream analysis.

## Single-cell RNA sequencing data analysis

Gene expression was analyzed using the Seurat package in R (*Hao et al., 2021*). The scaled TPM matrix and metadata information were used to create a Seurat object with the following specifications: min.cell=3, min.feature=200. Putative low-quality cells and doublets were removed based on the distribution of genes per cell (*Butler et al., 2018*). Cells were then checked for alignment to the human genome (GRCh38) and the fraction of mitochondrial genes as a measure of cell stress. Quality control criteria for cell exclusion were set as follows: percentage of mitochondrial genes >20%, percentage of aligned reads <60%, and the number of total genes expressed within 5000 and 10,000 (*Figure 1—figure supplement 1b and c*). A total of 276 cells passed quality control and did not affect the distribution of the four CD27/38 populations, indicating that all populations were of equal quality, and could be used for further analysis (*Figure 1—figure supplement 1d*). Gene expression values were then transformed to log2 TPM+1 and these values were used for all downstream analyses. Before performing dimensional reduction, the *FindVariableFeatures* function was applied to retrieve

the top 2000 high variable genes followed by a global data scaling using the *ScaleData* function. Principal component analysis (PCA) was chosen as a linear dimensionality reduction technique using the previously selected 2000 high variable genes as input and allowing 50 components. PCA-based cell clustering was performed by applying the *FindNeighbors* function on the first 10 PCAs followed by the *FindClusters* function with 0.5 resolution and the Leiden algorithm. For visualization purposes, a second nonlinear dimensionality reduction was performed using the *RunUMAP* function on the first 10 PCAs (*Figure 1—figure supplement 1e*).

### Differential expression analysis

Differential expression analysis was performed on all genes using the wilcoxauc method from *presto* (*Korsunsky et al., 2019*). GSEA was performed using the *clusterProfiler* package in R on 17,716 genes ranked based on logFC * adjusted p-value using the Gene Ontology (GO) and the hallmark human gene sets obtained from MsigDB4 (*Liberzon et al., 2015*). For visualization, hierarchical clustering of GSEA results was performed using the *pairwise_termsim* function from *enrichplot* with an adjusted p-value<0.05 and then plotted using the *emapplot* function.

### Publicly available (sc)RNAseq datasets

For the analysis of the publicly available scRNA-seq dataset of B cells from different human tissues (*Attaf et al., 2020*; *Jardine et al., 2021*; *Rizzetto et al., 2018*), processed gene expression matrixes were downloaded and cells were annotated as provided by the authors. For similarity and enrichment score analyses, the publicly available genes set GSE12366 that includes gene sets from up- and down-regulated genes found in sorted naive B cells, memory B cells, GC B cells, and plasma cells were used (*Longo et al., 2009*). The similarity score was then calculated using *AddModuleScore* from Seurat. For the analysis of similarity score over differentiation as reported in *Figure 4I*, the upregulated gene sets were compared and genes that were uniquely expressed by one population were used to calculate the similarity score to that population.

### Trajectory inference

Pseudotemporal ordering of the differentiating B cells trajectory, a PAGA (*Wolf et al., 2019*) was performed for the transcriptionally distinct cell clusters using the tl.paga function in *scanpy*. Velocity-based pseudotime reconstruction was performed using RNA velocity. To obtain the spliced and unspliced matrix for RNA velocity analysis, original fastq files were re-aligned using STAR (v2.7.7). Obtained.sam files were converted to sorted.bam files using *samtools* (v1.11). Sorted.bam files were fed into *velocyto* (v0.17.17) using the run-smartseq2 option. The obtained *loom* file containing the spliced and unspliced count matrix was imported into python and then merged with the Seurat object after conversion to *annData* format (*convertFormat* function from the *sceasy* R package). RNA velocity analysis was then performed using the *scVelo* (v0.2.2) package in python.

### B cell receptor analysis

Reconstruction of the rearranged B cell receptor sequences was performed using BraCeR (*Lindeman et al., 2018*) on the cells that passed quality control. The functions *assemble* and *summarise* were run to obtain respectively all possible BCR rearrangements and only the functional BCR rearrangements per cell. For every cell, one functional heavy chain and one functional light-chain BCR rearrangements were selected based on the highest expression. These sequences were then submitted to the HighV-quest portal of the International ImMunoGeneTics information system website (http://www.imgt.org) for further BCR characterization. Further analysis of the BCR rearrangements was performed using the tools from the Immcantation portal (https://immcantation.readthedocs.io/en/stable/). In particular, *Change-O* was used to perform clonal clustering and *Alakazam* was used for the analysis of somatic hypermutation.

## Acknowledgements

We thank the core facility at Sanquin, Erik Mul, Simon Tol, and Mark Hoogenboezem, for providing technical assistance. This research project was supported by a Landsteiner Foundation for Blood Transfusion Research grant (LSBR 1609) and a Product and process development grant for cellular

products by Sanquin (PPOC17-34/L2263). MB was supported by the German Research Foundation (IRTG2168-272482170, SFB1454-432325352).

## Additional information

### Funding

| Funder | Grant reference number | Author |
|---|---|---|
| Landsteiner Foundation for Blood Transfusion Research | LSBR 1609 | S Marieke van Ham |
| Sanquin Blood Supply Foundation | PPOC17-34/ L2263 | S Marieke van Ham |
| Deutsche Forschungsgemeinschaft | 272482170 | Marc Beyer |
| Deutsche Forschungsgemeinschaft | SFB1454-432325352 | Marc Beyer |

The funders had no role in study design, data collection and interpretation, or the decision to submit the work for publication.

### Author contributions

Niels JM Verstegen, Designed the experiments; Sabrina Pollastro, Designed the experiments; Peter-Paul A Unger, Designed the experiments; Casper Marsman, Performed the experiments; George Elias, Analyzed the data; Tineke Jorritsma, Performed the experiments; Marij Streutker, Performed the experiments; Kevin Bassler, Provided intellectual input to the study; Kristian Haendler, Provided intellectual input to the study; Theo Rispens, Provided intellectual input to the study; Joachim L Schultze, Provided intellectual input to the study; Anja ten Brinke, Supervised the study; Marc Beyer, Supervised the study; S Marieke van Ham, Conceived the study

### Author ORCIDs

Niels JM Verstegen ⓘ http://orcid.org/0000-0001-5732-4979
Sabrina Pollastro ⓘ http://orcid.org/0000-0002-0950-7583
George Elias ⓘ http://orcid.org/0000-0001-8419-9544
Kevin Bassler ⓘ http://orcid.org/0000-0002-4780-372X
S Marieke van Ham ⓘ http://orcid.org/0000-0003-1999-9494

### Ethics

All the healthy donors provided written informed consent following the protocol of the local institutional review board, the Medical Ethics Committee of Sanquin Blood Supply, and conforms to the principles of the Declaration of Helsinki.

### Decision letter and Author response

Decision letter https://doi.org/10.7554/eLife.83578.sa1
Author response https://doi.org/10.7554/eLife.83578.sa2

## Additional files

### Supplementary files
- MDAR checklist
- Supplementary file 1. Primer sequences.

### Data availability

All raw fastq files and digital gene expression matrixes (DGE) are available at the Gene Expression Omnibus (GEO) repository under accession no. GSE214265.

The following dataset was generated:

| Author(s) | Year | Dataset title | Dataset URL | Database and Identifier |
|---|---|---|---|---|
| Pollastro S, Unger PA, Marsman C, Elias G, Jorritsma T, Streutker M, Baβler K, Händler K, Rispens T, Schultze JL, ten Brinke A, Beyer M, van Ham SM, NJMC Verstegen | 2022 | Single-cell analysis reveals dynamics of human B cell differentiation and identifies novel B and antibody-secreting cell intermediates | https://www.ncbi.nlm.nih.gov/geo/query/acc.cgi?acc=GSE214265 | NCBI Gene Expression Omnibus, GSE214265 |

The following previously published datasets were used:

| Author(s) | Year | Dataset title | Dataset URL | Database and Identifier |
|---|---|---|---|---|
| Longo NS, Lugar PL, Yavuz S, Zhang W, Krijger PHL, Russ DE, Jima DD, Dave SS, Grammer AC, Lipsky PE | 2009 | Analysis of somatic hypermutation in X-linked hyper-IgM syndrome shows specific deficiencies in mutational targeting | https://www.ncbi.nlm.nih.gov/geo/query/acc.cgi?acc=GSE12366 | NCBI Gene Expression Omnibus, GSE12366 |
| Jardine L, Webb S, Goh I, Quiroga Londoño M, Reynolds G, Mather M, Olabi B, Stephenson E, Botting RA, Horsfall D, Engelbert J, Maunder D, Mende N, Murnane C, Dann E, McGrath J, King H, Kucinski I, Queen R, Carey CD, Shrubsole C, Poyner E, Acres M, Jones C, Ness T, Coulthard R, Elliott N, O'Byrne S, Haltalli MLR, Lawrence JE, Lisgo S, Balogh P, Meyer KB, Prigmore E, Ambridge K, Jain MS, Efremova M, Pickard K, Creasey T, Bacardit J, Henderson D, Coxhead J, Filby A, Hussain R, Dixon D, McDonald D, Popescu DM, Kowalczyk MS, Li B, Ashenberg O, Tabaka M, Dionne D, Tickle TL, Slyper M, Rozenblatt-Rosen O, Regev A, Behjati S, Laurenti E, Wilson NK, Roy A, Göttgens B, Roberts I, Teichmann SA, Haniffa M | 2021 | Blood and immune development in human fetal bone marrow and Down syndrome | https://data.humancellatlas.org/explore/projects/cc95ff89-2e68-4a08-a234-480eca21ce79 | Metadata and matrices, cc95ff89-2e68-4a08-a234-480eca21ce79 |
| Attaf N, Cervera-Marzal I, Dong C, Gil L, Renand A, Spinelli L, Milpied P | 2020 | FB5P-seq: FACS-based 5-prime end single-cell RNAseq for integrative analysis of transcriptome and antigen receptor repertoire in B and T cells | https://www.ncbi.nlm.nih.gov/geo/query/acc.cgi?acc=GSE137275 | NCBI Gene Expression Omnibus, GSE137275 |

*Continued*

| Author(s) | Year | Dataset title | Dataset URL | Database and Identifier |
|---|---|---|---|---|
| Rizzetto S, Koppstein DNP, Samir J, Singh M, Reed JH, Cai CH, Lloyd AR, Eltahla AA, Goodnow CC, Luciani F | 2018 | B-cell receptor reconstruction from single-cell RNA-seq with VDJPuzzle | https://bitbucket.org/kirbyvisp/vdjpuzzle2 | Metadata and matrices, vdjpuzzle2 |

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
