## [Editor Report]

To recapitulate B cell differentiation process in vitro, the authors established an in vitro system to identify a cluster and performed extensive analyses to demonstrate that the cluster mimics human germinal center antibody-secreting cells (ASCs). They provide stepwise trajectories of plasma cell differentiation from human naive B cells upon stimulation with CD40 ligands, IL-4, and IL-21. Since intermediate clusters of cells show features of germinal center B cells, the authors propose novel intermediate stages of B cells before a complete differentiation into plasma cells. This study is valuable in the differentiation of primary naive B cells into ASC ex vivo and may interest immunologists with an emphasis in B cell biology as it provides an in-depth description of the B cell differentiation pathways.

---

## [Decision Letter]

**Decision letter after peer review:**

Thank you for submitting your article "Single-cell analysis reveals dynamics of human B cell differentiation and identifies novel B and antibody-secreting cell intermediates" for consideration by *eLife*. Your article has been reviewed by 2 peer reviewers, including Murim Choi as the Senior Editor, Reviewing Editor and Reviewer #1.

Essential revisions:

The reviewers agreed with the value and novelty of the study, especially on not much studied B cell differentiation. But they also thought that substantial revisions are needed to guarantee publication in *eLife*. For example, the major concerns were

(1) Both reviewers showed concern on normalizing cell cycle effect would overfit the data.

(2) The degree their in vitro model to recapitulate in vivo situation.

(3) Description on methods for replication in other groups, and difference from the previous paper (Unger 2021).

*Reviewer #2 (Recommendations for the authors):*

This work is interesting and provides an in-depth description of the B cell differentiation pathways. However, this work has several pitfalls, which may decrease the value of this manuscript.

1. This study uses in vitro system of germinal center B or plasma cell differentiation from human naive B cells. Even though cells of clusters 4 and 5 express the ASC genes also found in vivo, these in vitro generated cells may still be different from in vivo plasma cells in other gene expression profiles determined by in vivo conditions. Therefore, a novel pre-ASC population could be found only in vitro. The in vitro B cell culture system is suitable for the investigation of molecular differentiation mechanism but may be not ideal for the description of differentiation trajectory or pathway.

2. The authors removed the influence of the cell cycle by performing cell cycle regression, which is good for removing the effects of the genes oscillating with the cell cycle. But, the cell division numbers may affect the choice between memory B cell and plasma cell differentiation [Zhou, Front Immunol, 9:2053; Duffy, Science, 335:338]. More ASCs may appear after multiple rounds of cell division stochastically. Therefore, it would be nice to follow gene expression profiles with cell division numbers or different time points of culture.

3. IL-21 and IL-4 are principal cytokines for germinal center reaction and CD40/CD40L interaction is essential for GC formation. So the culture condition appears to favor the GC-like cells initially, and ASCs may appear late after multiple rounds of cell division. Therefore, the author's argument of the first identification of GC-like B cells may be regarded as a case of the human counterpart of GC B cell culture of mice [Gonzalez, J Immunol, 201:3569].

---

## [Author Response]

Reviewer #2 (Recommendations for the authors):This work is interesting and provides an in-depth description of the B cell differentiation pathways. However, this work has several pitfalls, which may decrease the value of this manuscript.1. This study uses in vitro system of germinal center B or plasma cell differentiation from human naive B cells. Even though cells of clusters 4 and 5 express the ASC genes also found in vivo, these in vitro generated cells may still be different from in vivo plasma cells in other gene expression profiles determined by in vivo conditions. Therefore, a novel pre-ASC population could be found only in vitro. The in vitro B cell culture system is suitable for the investigation of molecular differentiation mechanism but may be not ideal for the description of differentiation trajectory or pathway.

We appreciate the insight provided by the reviewer regarding the potential differences between in vitro and in vivo generated ASCs and now specifically addressed this point in our manuscript: We observed that a significant portion of the differentially expressed genes in our in vitro ASCs are not shared with ex vivo ASC populations. Specifically, 293 out of 1406 genes (20%) identified in clusters 4 and 5 were unique to our in vitro ASCs (Figure 2h). Additionally, we have conducted additional comparisons and found that ex vivo ASC populations also possess unique differentially expressed genes that are not shared with other ex vivo *or* in vitro*-generated* ASC populations (Figure 2i). These findings indicate that in vivo ASCs carry specific, possibly tissue-related, gene signatures in addition to the "core ASC signature" shared among ASCs. We have added this in the current manuscript (line 360-363).

In regard to the question of whether the pre-ASC cell population observed in our in vitro system is in fact present in vivo, we agree with the reviewer that we did not prove the presence of the same population in vivo setting. However, we did show transcriptional overlap between cells in cluster 4 and small population of B cell in ex-vivo obtained tissues with active ASC formation (Figure3h). Additionally, in the recently published study by Sanz *et al.,* Front. Immunol. the authors identified a pre-ASC population in the peripheral blood of healthy volunteers six days post-vaccination. These cells displayed intermediate CD27 and CD38 expression, lower expression of PRDM1, and higher expression of PAX5 compared to CD138 plasma blasts, which are consistent with the characteristics observed in cluster 4 of our study (Figure 1—figure supplement 2 and Figure 1l-m). Therefore, we believe that our in vitro pre-ASC population is likely the same as the one found in vivo, however, it may be challenging to identify in normal steady-state conditions as it only appears transiently during ASC formation. We have rephrased our conclusion according and added the aforementioned speculation in the discussion (lines 471-477).

2. The authors removed the influence of the cell cycle by performing cell cycle regression, which is good for removing the effects of the genes oscillating with the cell cycle. But, the cell division numbers may affect the choice between memory B cell and plasma cell differentiation [Zhou, Front Immunol, 9:2053; Duffy, Science, 335:338]. More ASCs may appear after multiple rounds of cell division stochastically. Therefore, it would be nice to follow gene expression profiles with cell division numbers or different time points of culture.

In regards to the concerns about the cell-cycle regression step in our bioinformatic pipeline, we fully agree with the reviewer and know that the cell cycle plays a significant role in B cell differentiation output trajectories. Preparing the manuscript, we have in fact performed a parallel analysis in which we compared both cell cycle regressed- and not cell cycle regressed-based clustering and marker gene selection. Concerning the clustering, other clusters were obtained using the not cell-cycle-regressed dataset compared to the cell-cycle-regressed dataset (figure below). However, when overlaying the clusters obtained with the cell cycle-regressed dataset, the extra clusters were the same cell population but now split based on cycling and not cycling cells: cluster 2 is now divided into the cycling cluster “c”, and the not-cycling cluster “d” while cluster 4 and 5 are now divided into the cycling clusters “e” and the not-cycling cluster “f”. A comprehensive examination of the expression of the top 50 genes associated with antibody-secreting cells in the (non)cycling clusters 4 and 5 reveals that these genes are expressed at a higher level in (non)cycling cluster 5 as compared to cluster 4. This suggests that the cells within cluster 5 are more advanced in their differentiation, regardless of their cell cycle state. This finding has led us to the decision to present the data that has undergone cell cycle regression in the manuscript. Should the reviewer so desire, we are very willing to include additional figures in the manuscript that include the un-regressed representation.

With regards to the suggestion of further investigating the effect of cell division numbers on differentiation pathway choice, we have already conducted research in this area and have recently published our findings (Marsman *et al.*, Cells 2020). We recognize the value in further exploring this topic by conducting a single-cell analysis of the entire culture system at different time points and incorporating a proliferation dye in the starting cell population. As these conditions were not included in the present study, we are unable to provide further insight at this time. However, we are currently in the process of setting up such experiments and hope to present the results in the near future.

3. IL-21 and IL-4 are principal cytokines for germinal center reaction and CD40/CD40L interaction is essential for GC formation. So the culture condition appears to favor the GC-like cells initially, and ASCs may appear late after multiple rounds of cell division. Therefore, the author's argument of the first identification of GC-like B cells may be regarded as a case of the human counterpart of GC B cell culture of mice [Gonzalez, J Immunol, 201:3569].

We thank the reviewer for the point raised. We are in total agreement, while a similar culture system is well-established and commonly used in mouse B cell biology, studies utilizing human B cells are limited in comparison. To the best of our knowledge, our study represents the first comprehensive analysis of such a system in humans. We have now added another comparative sentence to the mice studies in our manuscript (lines 481-483).